# The Shifting Mycotoxin Profiles of Endophytic *Fusarium* Strains: A Case Study

Gelsomina Manganiello [1], Roberta Marra [1], Alessia Staropoli [1,2], Nadia Lombardi [1], Francesco Vinale [1,2] and Rosario Nicoletti [1,3,*]

[1] Department of Agricultural Sciences, University of Naples Federico II, 80055 Portici, Italy
[2] Institute for Sustainable Plant Protection (IPSP), National Research Council (CNR), 80055 Portici, Italy
[3] Council for Agricultural Research and Economics, Research Centre for Olive, Citrus and Tree Fruit, 81100 Caserta, Italy
* Correspondence: rosario.nicoletti@crea.gov.it

**Abstract:** *Fusarium* species are known to establish manifold interactions with wild and crop plants ranging from pathogenicity to endophytism. One of the key factors involved in the regulation of such relationships is represented by the production of secondary metabolites. These include several mycotoxins, which can accumulate in foodstuffs causing severe health problems to humans and animals. In the present study, an endophytic isolate (A1021B), preliminarily ascribed to the *Fusarium incarnatum-equiseti* species complex (FIESC), was subjected to biochemical and molecular characterization. The metabolomic analysis of axenic cultures of A1021B detected up to 206 compounds, whose production was significantly affected by the medium composition. Among the most representative products, fusaric acid (FA), its derivatives fusarinol and 9,10-dehydro-FA, culmorin and bikaverin were detected. These results were in contrast with previous assessments reporting FIESC members as trichothecene rather than FA producers. However, molecular analysis provided a conclusive indication that A1021B actually belongs to the species *Fusarium babinda*. These findings highlight the importance of phylogenetic analyses of *Fusarium* species to avoid misleading identifications, and the opportunity to extend databases with the outcome of metabolomic investigations of strains from natural contexts. The possible contribution of endophytic strains in the differentiation of lineages with an uneven mycotoxin assortment is discussed in view of its ensuing impact on crop productions.

**Keywords:** endophytic fungi; *Fusarium*; species complexes; mycotoxins; fusaric acid; trichothecenes; biosynthetic gene clusters

## 1. Introduction

*Fusarium* species have been commonly reported in the majority of bioclimatic regions and ecosystems, where they occur as endophytes, latent plant pathogens, or soil saprobes, thus showing a considerable ecological plasticity [1–3]. Some *Fusarium* species may cause severe plant diseases, and contaminate crop productions with mycotoxins, which are secondary metabolites (SMs) of major concern to food and feed safety worldwide [4–7]. This aptitude may not only involve pre- and post-harvest plant pathogens, but also strains which develop endophytically without causing disease symptoms [8,9]. On the other hand, the release of bioactive SMs in plant tissues by endophytic strains may induce defensive responses against pests and pathogens [10], with positive implications on plant growth [11].

The genus *Fusarium* is also well known for a controversial taxonomic history, where species descriptions were basically founded on key morphological characters [3]. In the last decades several

studies considered data on SM production as a possible support in *Fusarium* taxonomy [12–14]. However, such a sound approach has been impaired by the finding that synapomorphy, i.e., the occurrence of certain common characters in distantly related organisms, has notably affected the classification of *Fusarium* strains in the past [15]. More recently, the advances in the DNA-sequencing technology allowed the identification of *Fusarium* spp. based on multi-gene genealogies [16–19], thus improving the phylogenetic accuracy and the taxonomic resolution [20–22]. Nevertheless, the ongoing deposition of DNA sequences in database resulting from the manifold surveys of natural populations of *Fusarium*, together with the characterization of novel species, may result in incorrect matches and, subsequently, in misleading identifications [23].

In this work, we report a case study describing a *Fusarium* strain (A1021B) that was recovered as an endophyte of common spindle (*Euonymus europaeus*) at the Astroni Natural Reserve near Naples, Italy. The fungus was provisionally defined to belong to the *Fusarium incarnatum-equiseti* species complex (FIESC) [24]. Fusarinol, a derivative of fusaric acid (FA), was the main extrolite purified from cultures grown in Czapek-Dox broth (CDB) [25]. Afterwards, we found FA to be the major SM produced by A1021B in potato dextrose broth (PDB), thus confirming that FA production in vitro is influenced by the culture medium composition [26–28]. Previous studies reported that FA production is strain-dependent even in species known as common producers, and it can be stimulated in some reluctant *Fusarium* strains by co-cultivation with other fungi [29]. Factors regulating gene expression are fundamental in explaining variation in SM production. Gene clusters for FA synthesis have been detected in many *Fusarium* spp. [15,30], and the deletion of specific genes has been reported to affect the production of FA and related compounds [31,32].

Our results appeared in contrast with the mycotoxin profile commonly associated with FIESC members. In fact, previous investigations failed to detect the production of FA in species/strains ascribed to this species complex [26,33–35], which are mainly known as trichothecene producers [35,36]. FA was listed among the mycotoxins of *F. equiseti* in a couple of recent papers [15,28], but no specific references were provided supporting this inference. Therefore, further investigations were undertaken concerning both the authentic taxonomic identity of strain A1021B and its biochemical and molecular characterization.

## 2. Materials and Methods

### 2.1. Fungal Strain and Culture Conditions

The *Fusarium* strain A1021B was maintained on potato dextrose agar (PDA, HI-Media, Mumbai, India) at 4 °C, and subcultured bimonthly. For metabolomic investigation CDB and PDB (both from HI-Media) cultures were prepared in 250-mL flasks containing 100 mL of broth. Twelve flasks were prepared for each medium and inoculated with 10 plugs (5 mm diameter) from 6-day old PDA cultures. Six flasks were incubated at 25 °C in a growth chamber with 16:8 h photoperiod, while the remaining 6 were incubated at 25 °C in darkness. These batches were further divided into two groups (each including three replicates) which were grown for one or two weeks, respectively. Then fungal debris were filtered through three layers of cheesecloth, and the filtrates were stored at −20 °C.

Solid fermentation of strain A1021B was carried out on maize kernels (MK). After rinsing three times in sterile water, 30 g of kernels were placed in each of twelve 250-mL flasks and sterilized (121 °C, 20 min). Five milliliters of sterile water were added to each flask, that were subsequently inoculated as described for liquid cultures. Similarly, the flasks were grouped in 4 batches, each consisting of three replicates, and incubated at 25 °C as described above. After one or two weeks, a 10 g sample was taken from each MK flask and separately ground to be further processed.

## 2.2. Culture Extraction and LC-MS Analysis

MK samples were extracted in 8 mL of 50% methanol in water. Samples were centrifuged (10 min at 16,100 g, 4 °C), and the supernatants were collected. These, as well as samples from liquid cultures, were filtered through 0.2 μm polyvinylidene fluoride filters (Chromacol, Welwyn Garden City, UK).

SM profiling was carried out through a 6540 Ultra High Definition (UHD) Accurate Quadrupole Time-of-Flight (Q-TOF) Liquid Chromatography tandem Mass-Spectrometry (LC-MS/MS) mass spectrometer (Agilent Technologies, Santa Clara, CA, USA) with a Dual Electrospray Ionization (ESI) source, coupled to a 1200 series Rapid Resolution High Performance Liquid Chromatography (HPLC) with a Diode Array Detector (DAD) system (all from Agilent Technologies). Samples (7 μL) were injected onto a Poroshell 120EC-C18 1.8 pm, 2.1 × 5 mm reverse phase analytical column (Agilent Technologies) at a constant temperature (35 °C). Mobile phases consisted of (A) water (Cromasolv® Plus, LC-MS-Sigma) and (B) acetonitrile (Cromasolv® Plus, LC-MS-Sigma) both acidified with 0.1% LC-MS grade formic acid. The analyses were carried out at a flow rate of 0.6 mL min$^{-1}$ with the following gradient: 0 min—5% B; 12 min—100% B; 15 min—100% B; 17 min—95% B; 20 min—95% B; 2 min post-time. The UV spectra were collected by DAD every 0.4 s from 190 to 750 nm with a resolution of 2 nm. The source conditions for electrospray ionization were the following: nitrogen gas temperature was 350 °C with a drying gas flow rate of 11 L min$^{-1}$ and a nebulizer pressure of 45 psig. The fragmentor voltage was 180 V and skimmer voltage 45 V. The range acquisition of TOF spectra was from 50 to 1600 m/z with an acquisition rate value of 3 spectra s$^{-1}$. Data were collected in positive ion mode. The real-time lock mass correction was performed by using two reference mass solutions including purine ($C_5H_4N_4$ at m/z 121.050873, 10 μmol L$^{-1}$) and hexakis (1H,1H,3H-tetrafluoropentoxy)-phosphazene ($C_{18}H_{18}O_6N_3P_3F_{24}$ at m/z 922.009798, 2 μmol L$^{-1}$). These solutions were purchased from Agilent Technologies and injected into MS by an isocratic pump at a constant flow rate (0.06 mL min$^{-1}$). Solvents were LC–MS grade, and all other chemicals were analytical grade. All were from Sigma-Aldrich (Steinheim, Germany) unless otherwise stated.

Mass spectra were analyzed through the MassHunter Qualitative Analysis Software B.06.00 (Agilent Technologies), and then through the MassProfile Professional Software (Agilent Technologies) to compute the annotation and statistical analyses. LC-MS data were compared to known compounds included in an in-house database, as previously described [37,38].

Graphical representations were performed using ClustVis, a web-based multivariate data analysis tool. The principal component analysis (PCA) was performed using the Singular Value Decomposition (SVD) with imputation algorithm in ClustVis online tool. Data on SMs were summarized using the heatmap function in ClustVis tool with row centered and unit variance scaling applied. The hierarchical clustering was obtained using correlation method. Compounds with normalized intensity values >2 were used to analyze common and unique entities in the different treatments by Venn diagrams with the online tool jvenn (http://jvenn.toulouse.inra.fr/app/index.html).

## 2.3. DNA Extraction and PCR Conditions

Isolate A1021B was grown in PDB on a rotary shaker at 120 rpm for 96 h at 25 °C. Fresh mycelium was collected after vacuum filtration through No. 4 Whatman filter paper (Whatman Biosystems Ltd., Maidstone, UK), then frozen in liquid nitrogen, ground to a fine powder and stored at −80 °C until further processing. Total genomic DNA was extracted from 10 mg of ground mycelium by using the NucleoSpin® Soil kit (Macherey-Nagel, Düren, Germany) according to the manufacturer's protocol. Sequences of the housekeeping genes calmodulin (*CAL1*), translation elongation factor (*TEF1*), β-tubulin (*TUB2*) and internal transcribed spacer 1–4 (ITS) were amplified using the following PCR program: denaturation at 96 °C for 2 min; 35 cycles of denaturation at 94 °C for 30 s, annealing at 55 °C for 30 s; extension at 68 °C for 75 s; and final extension at 68 °C for 10 min. Before sequencing, PCR products were purified using PureLink PCR purification kit (Invitrogen, Paisley, UK) following the manufacturer's instructions. Furthermore, the presence of amplicons related to the trichothecene

biosynthetic genes *TRI1*, *TRI4*, *TRI5*, *TRI8*, *TRI11* was investigated. All primers used in this work are reported in Table 1.

**Table 1.** Primers used in this study for DNA sequence amplifications.

| Target Gene | 5′–3′ Sequence | References |
|:---:|:---:|:---:|
| *TRI1* | GCGTCTCAGCTTCATCAAGGCAKCKAMTGAWTCG CTTGACTTSMTTGGCKGCAAAGAARCGACCA | [39] |
| *TRI4* | CCAATCAGYCAYGCTRTTGGGATACTG ACCCGGATTTCRCCAACATGCT | [39] |
| *TRI5* | GGCATGGTCGTGTACTCTTGGGTCAAGGT GCCTGMYCAWAGAAYTTGCRGAACTT | [39] |
| *TRI8* | GACCAGNAYCACSGYCAACAGTTCAG GAACAGCCRCTCCRWAACTATTGTC | [35] |
| *TRI11* | TWCCCCACAAGRAACAYCTYGARCT TCCCASACTGTYCTSGCMAGCATCAT | [35] |
| *CAL* | GARTWCAAGGAGGCCTTCTC TTTTGCATCATGAGTTGGAC | [17] |
| *TEF1* | ATGGGTAAGGARGACAAGAC GGARGTACCAGTSATCATGTT | [16] |
| *TUB2* | GGTAACCAAATCGGTGCT ACCCTCAGTGTAGTGACCCTYTGGC | [40] |
| ITS 1–4 | CTTGGTCATTTAGAGGAAGTAA TCCTCCGCTTATTGATATGC | |

## 2.4. Species Identification and Phylogenetic Analysis

Phylogenetic relationships of strain A1021B were investigated on account of *CAL1*, *TEF1* and *TUB2* sequences as reported in [15]. DNA sequences were blasted against the NCBI GenBank database using default parameters and then aligned with isolates belonging to the FIESC [15,41] and *Fusarium babinda* by the Clustal W algorithm [42] with MEGA7 software [43]. Phylogenetic trees were inferred using the maximum likelihood method based on Tamura-Nei model applied to the whole set of manually edited aligned sequences. The confidence of the branching was estimated by bootstrap (BP) analysis (1000 BP). A strain of *Fusarium concolor* was used as outgroup for rooting the phylogenetic tree. DNA sequences of the three loci were submitted to GenBank, with the following accession numbers: MK968883 (ITS), MK984207 (*CAL1*), MK984206 (*TEF1*), and MK984208 (*TUB2*).

## 3. Results

### 3.1. Metabolome Analysis

The investigation on SM production in liquid (CDB, PDB) or solid (MK) media, the latter representing a commonly used substrate to evaluate mycotoxin production in *Fusarium* spp. [28], revealed that up to 206 compounds are synthesized by strain A1021B in axenic cultures. The PCA score plot demonstrated a differential and significant effect of the medium composition on SM production (Figure 1A). Moreover, the assortment of SMs produced in CDB or PDB was less affected by light exposure than by the culturing time (1 vs. 2 weeks). On the other hand, on MK the SM profile was particularly influenced by the former factor, i.e., multiple compounds after one week of growth in darkness were produced. To obtain a simplified representation of the different assortments, a heatmap clustering compounds was generated (Figure 1B) by selecting 29 entities which made it possible to discriminate among the different treatments, selected on PCA. Among them, FA (179.0974 Da) and its derivatives fusarinol (165.1181 Da) and 9,10-dehydro-FA (177.0799 Da) were detected. Other putatively identified SMs were bikaverin (382.1126 Da), a tetracyclic benzoxanthone whose genetic base is reported to be clustered to FA [44,45], and culmorin (238.1446 Da), a sesquiterpenoid which is often associated with trichothecene production [46,47].

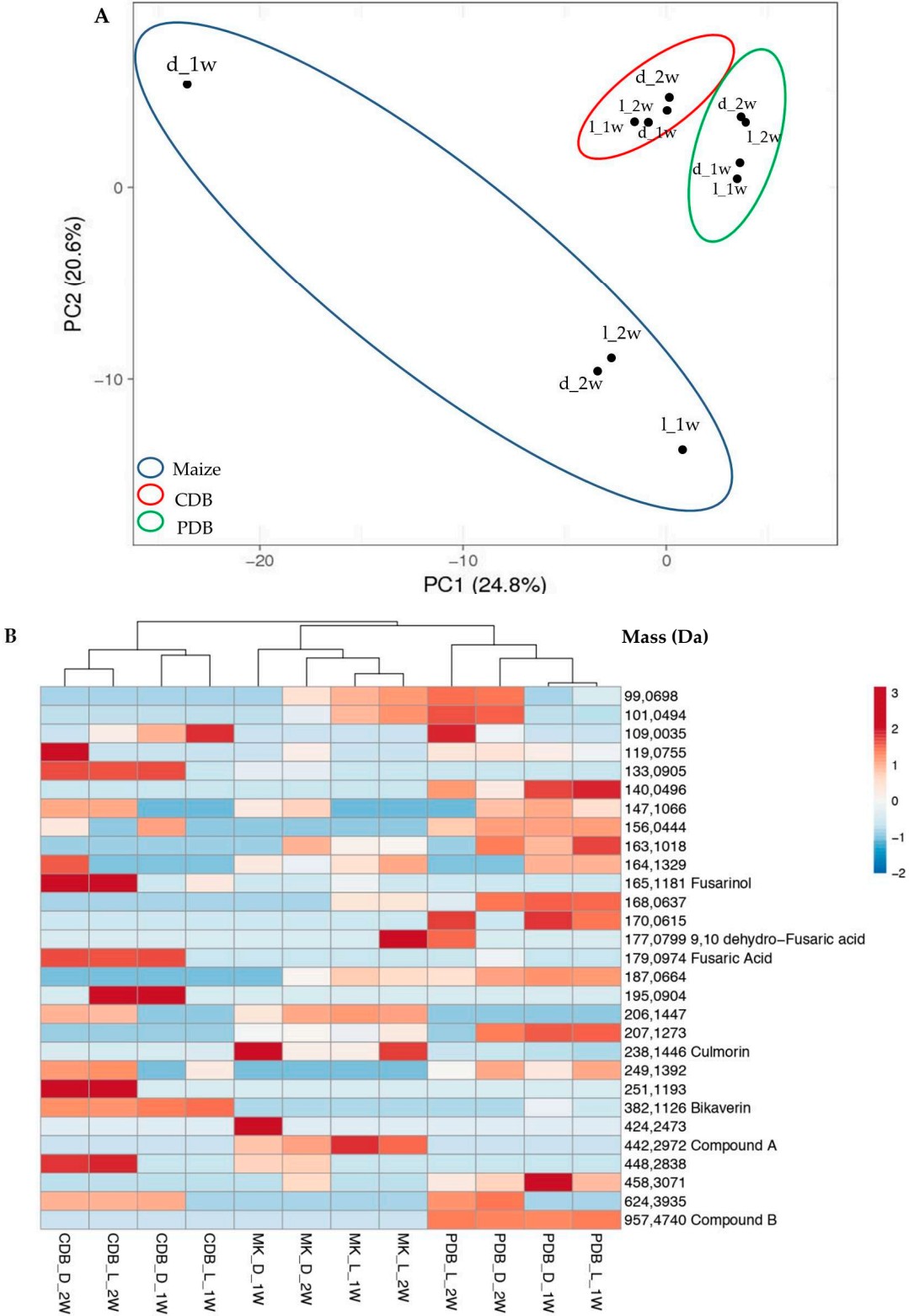

**Figure 1.** (**A**) Principal component analysis (PCA) score plot of secondary metabolites (SMs) produced by A1021B under different growth conditions. (**B**) Heat map illustrating the abundance of the main SMs in A1021B cultures, visualized through the color scale reported on the right. Each row represents differentially abundant products ordered by their mass (Da), while columns correspond to the different culturing conditions. MK = maize kernels; CDB = Czapek-Dox broth; PDB = potato dextrose broth; d = darkness; l = light; 1 w = 1 week; 2 w = 2 weeks.

Among the identified compounds, the LC-MS Q-TOF analysis revealed that fusarinol was predominantly produced in CDB, in dark as well as in light conditions, while 9,10-dehydro-FA accumulated in PDB and MK. However, both compounds were found only after two weeks of growth. The production of FA was mostly observed in CDB maintained in darkness, or in light exposure after two weeks only. A similar biosynthetic course is displayed by culmorin in PDB, while the production of bikaverin was mainly detected in CDB cultures regardless to the presence/absence of light. Furthermore, among the unidentified molecules, compound A and compound B (Figure 1B) were particularly affected by medium composition. In fact, compound A was detected exclusively in MK while compound B was produced only in PDB, and their production was not related to specific growth condition (1–2 w; light/darkness).

Our analysis did not show the production by A1021B of 8-O-methylbostrycoidin, a polyketide pigment which has been reported in association with FA [48]. Furthermore, no trichothecenes were detected in any cultivation condition.

Venn diagrams showed that A1021B was able to synthesize specific compounds in the different media, and that only a small part of them was in common among the three conditions (Figure 2). In darkness, the growth on MK enhanced the production of specific SMs at both time points considered (81 and 75 compounds, respectively, after 1 or 2 weeks of growth), while CDB was the least inductive medium (6 and 11 compounds, respectively). Moreover, very few compounds (3 and 4, respectively, after 1 or 2 weeks of growth) accumulated constitutively in darkness regardless to the medium. A similar distribution was observed when A1021B was cultivated under light exposure. In fact, MK represented the most inductive substrate, while in CDB few compounds accumulated. No specific SMs were detected in CDB at the first time point. Overall, SM production was higher in darkness (Figure 2), and MK was more effective in enhancing the production of certain compounds.

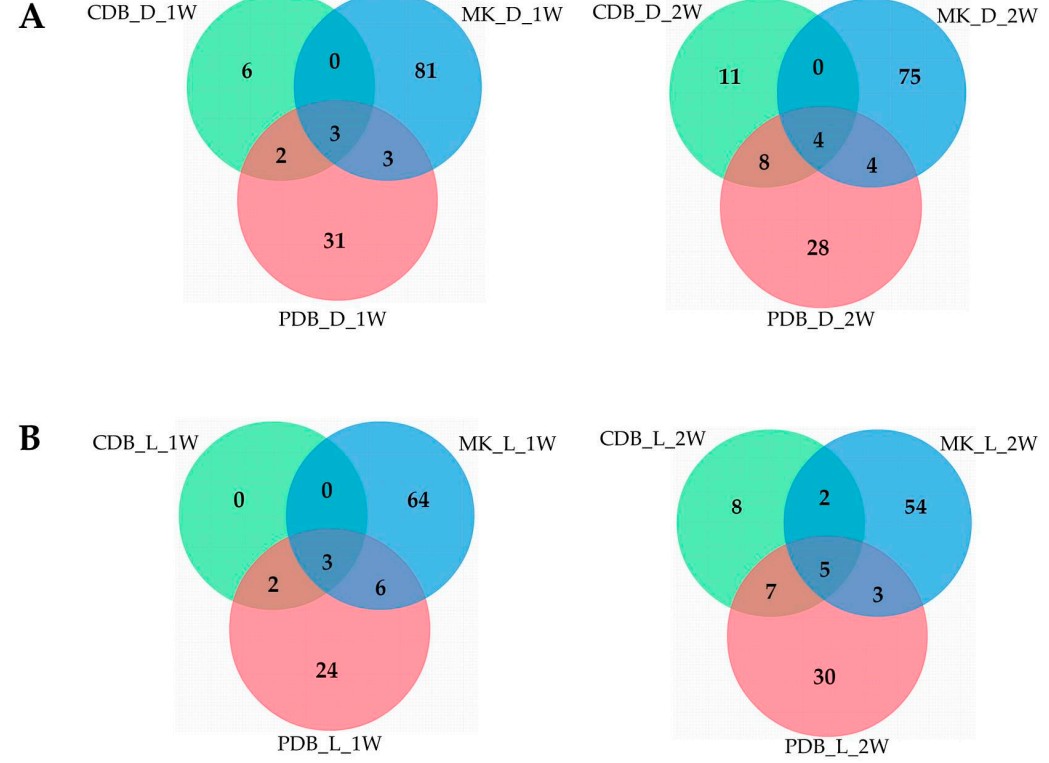

**Figure 2.** Venn diagrams showing the number of unique and overlapping products in A1021B cultures under the different growth conditions. (**A**) Secondary metabolites (SMs) produced in darkness (d) after one (1 w) or two weeks (2 w). (**B**) SMs produced under light exposure (l) after one (1 w) or two weeks (2 w). MK = maize kernels; CDB = Czapek-Dox broth; PDB = potato dextrose broth.

### 3.2. Genetic and Phylogenetic Analysis, and Species Identification

As trichothecenes may well characterize the mycotoxin profile of FIESC members, the presence in A1021B of genes involved in biosynthesis of these compounds was investigated by PCR as previously described [15]. Amplicons of all the selected regions (*TRI-1*, *TRI-4*, *TRI-5*, *TRI-8*, *TRI-11*) were detected (data not shown), indicating that strain A1021B actually holds the genetic features to produce these mycotoxins. Nevertheless, the related SMs were not detected in any of the culture conditions used in this study.

Even if the genetic data matched with the hypothesis that A1021B might belong to the FIESC, a different indication resulted from the phylogenetic analysis, conducted using concatamers of ITS, *TEF1* and *CAL1* sequences previously employed in the characterization of this species complex. In this experiment, a strain of *F. concolor* was used as the outgroup [41]. Interestingly, A1021B clustered with the latter instead of any of the several identified or unidentified FIESC members (Figure 3). Nevertheless, a new BLAST search in the NCBI database based on *TEF1* sequences did not yield a consistent homology with the available strains of *F. concolor*.

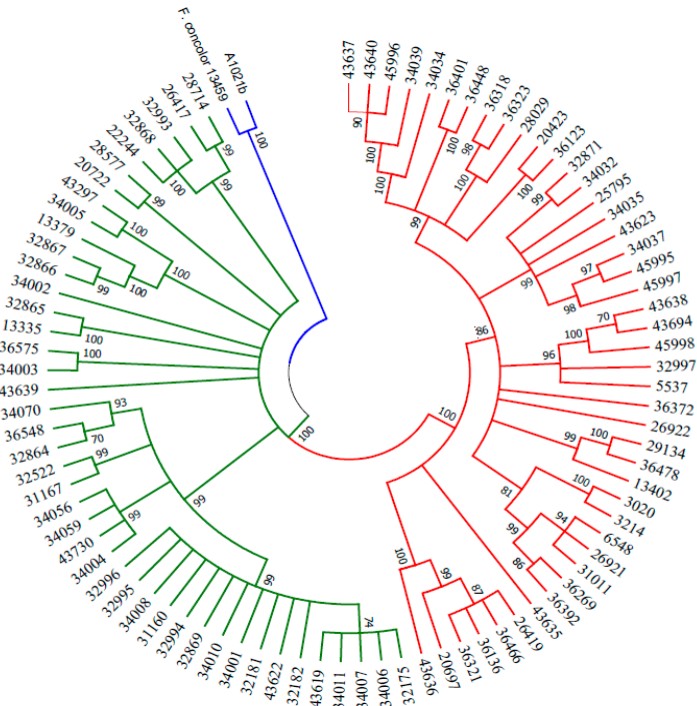

**Figure 3.** Maximum likelihood tree inferred from ITS-TEF1-CAL1 concatamers. Phylogenetic analysis including A1021B, FIESC members and *F. concolor* as outgroup inferred using the maximum likelihood method (MEGA7). The bootstrap consensus tree inferred from 1000 replicates is taken to represent the evolutionary history of the taxa analyzed.

The hypothesis that strain A1021B represented a novel taxon could explain such discrepancy. However, a subsequent BLAST search carried out in December 2018 revealed an unexpected 100% homology with a series of *TEF1* sequences from the species *F. babinda* [49], which were made available in October 2018 after another notable taxonomic revision [45]. Following this finding, another phylogenetic tree including isolates of *F. concolor*, *F. babinda* and FIESC was generated where A1021B clearly clustered with the strains of *F. babinda* (Figure 4).

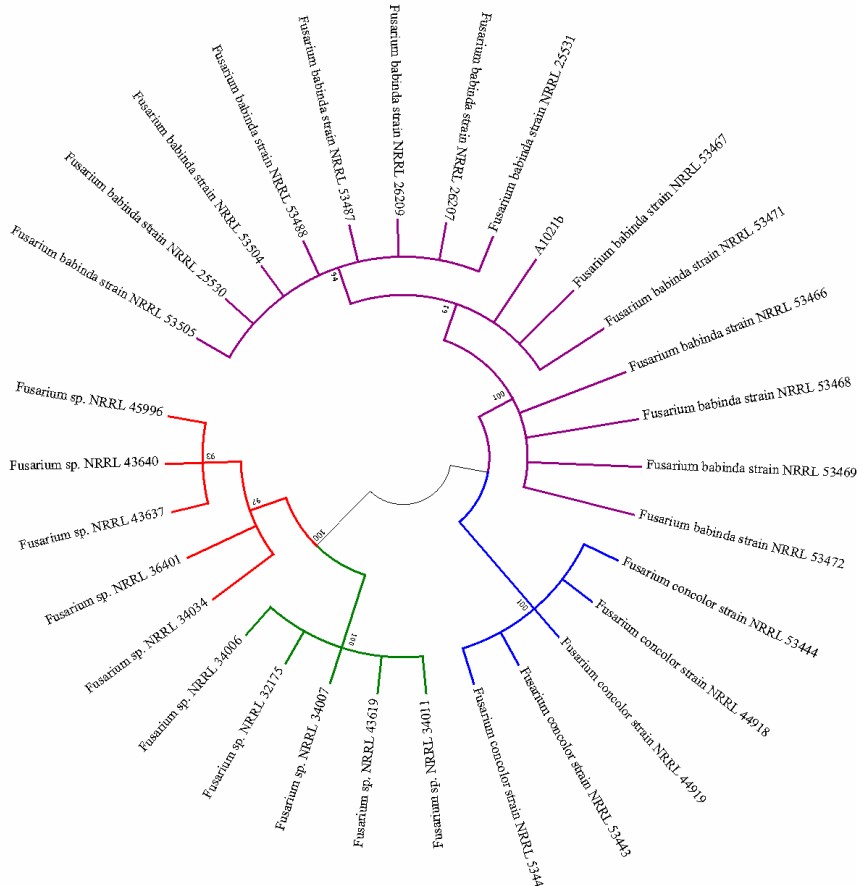

**Figure 4.** Maximum likelihood tree inferred from TEF1. Phylogenetic analysis including A1021B, strains of FIESC, *F. babinda* and *F. concolor* inferred using the maximum likelihood method (MEGA7). The bootstrap consensus tree inferred from 1000 replicates is taken to represent the evolutionary history of the taxa analyzed.

## 4. Discussion

A recent study [45] demonstrated that *F. babinda* represents the correct identification for many strains previously ascribed to *F. polyphialidicum*. This species, on the other hand, has now been reported as a synonym of *F. concolor*, deserving priority in taxonomy as an older accepted species name [50]. While *F. concolor/polyphialidicum* is known as a typical trichothecene producer [28,45], the mycotoxin profile of *F. babinda* seems to be centered on FA, and no clues of trichothecene biosynthetic abilities were detected in the limited assessments carried out so far. An analysis concerning the genetic basis for trichothecene synthesis in a single strain of this species (NRRL 25539) also provided negative results [45]. The same study reported that strain NRRL 25539 has the gene clusters for the production of some compounds (enniatins, fusarin, fusarubin), which, however, were absent in our cultures.

In this work, metabolomic analysis confirmed that in axenic cultures strain A1021B basically produced FA and some known compounds. Interestingly, bikaverin was found to accumulate mainly in CDB cultures, where the carbon source is represented by sucrose, in consistency with a previous report that the availability of this sugar stimulates bikaverin production in vitro [51]. Molecular data indicated the presence of trichothecene biosynthetic gene clusters, but they were not expressed under the culture conditions we tested, thus making A1021B divergent from strain NRRL 25539 [45]. Considering that *F. babinda*, which formerly had been reported only from Australia, turned out to have a worldwide diffusion [45], and that *F. polyphialidicum* was described as a typical producer of type-A trichothecenes [28], our finding highlights the need for more exhaustive investigations on the mycotoxin profile of this emerging species. In this respect, an assessment concerning occurrence of

*TRI-5* in *F. equiseti* detected this gene sequence in 50% of the examined strains only [36], confirming previous evidence of uneven production of trichothecenes in this species [52].

Recent evaluations of the mycotoxin-producing ability indicate that *Fusarium* phylogenetic relationship may vary, and non-conforming strains, new species or lineages often result after the exploration of new ecological contexts, particularly those involving endophytic fungi [53–55]. In fact, an intriguing ability to synthesize unexpected SMs can be ascribed to endophytes, which are able to establish physical contacts and eventually interact through horizontal gene transfer (HGT) with both plants and other microorganisms living in this particular ecological niche [56,57]. Indeed, ecological proximity has been considered to favor HGT [57].

In fungi, genes coding for the synthesis of SMs are typically adjacent to one another in clusters of co-expressed genes, including a core gene responsible for the synthesis of a basic structure, and side genes which control chemical modifications, transport, and regulation [58]. Biosynthesis of FA, bikaverin, culmorin and trichothecenes is governed by polyketide synthases, large multi-domain enzymes that catalyze sequential condensation of simple carboxylic acids. A few hundreds of gene sequences involved in the biosynthesis of polyketides have been detected in *Fusarium* spp., which corresponded to 67 clades in a phylogenetic analysis, where each clade refers to distinct products. This analysis also pointed out a genetic potential to synthesize compounds which are the same or similar to those known to be produced from other fungi, but not reported in *Fusarium* so far [59].

From an evolutionary viewpoint, HGT of gene clusters regulating mycotoxin biosynthesis is theoretically supported by the reasonable inference that clustering confers a selective advantage to the cluster itself [56,60]. In addition, the hypothesis that the *TRI-5* gene cluster may have spread among unrelated fungal species through HGT has already been advanced in the past [61]. HGT was also indicated as the means of transmission of a 5-gene cluster presiding over the synthesis of bikaverin from *Fusarium* to *Botrytis cinerea* [51], and as a more general evolutionary mechanism in *Fusarium* [62]. Moreover, it has been demonstrated that transfer of lineage-specific genomic regions occurred in *Fusarium*, including even entire chromosomes up to more than one-quarter of the genome, and involving genes related to pathogenicity. These were effective in converting pathogenic strains into non-pathogenic ones, and were possibly responsible for the emergence of new pathogenic lineages [63]. Therefore, natural ecosystems are recognized to play a role as reservoirs of novel crop pathogens with a meaningful impact on disease management and biosecurity [64].

## 5. Conclusions

In this work, we reported a case-study investigating the taxonomy and SM production in the endophytic *Fusarium* strain A1021B. As a consequence of the ongoing updates in the phylogenetic relationships of *Fusarium* species, the analyses of mycotoxin profile and selected gene sequences lead us to identify this isolate as *F. babinda*. Our findings support previous observations that SM production in axenic cultures by *Fusarium* strains does not necessarily conform to genetically based analyses, and that this limitation could be overcome in vivo where interaction with the host plant or other endophytic microorganisms may result in the activation of silent genes.

Besides sequences deposited in GenBank, strain A1021B is available on request for inclusion in phylogenetic and metabolomic studies.

**Author Contributions:** Conceptualization, R.N. and F.V.; methodology, G.M., R.M., A.S., N.L. and F.V.; writing—review and editing, G.M., R.M., F.V. and R.N.

**Funding:** The research activity of F.V., A.S., R.M. and G.M. was funded by MIURPON [grant number Linfa 03PE_00026_1; grant number Marea 03PE_00106]; MIUR-GPS [grant number Sicura DM29156]; POR FESR CAMPANIA 2014/2020- O.S. 1.1 [grant number Bioagro 559]; MISE [grant number Protection F/050421/01-03/X32].

**Conflicts of Interest:** The authors declare no conflict of interest.

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
