# Peer review of "The Shifting Mycotoxin Profiles of Endophytic Fusarium Strains: A Case Study"

_agriculture, doi:10.3390/agriculture9070143_

Round 1

Reviewer 1 Report

The article by Manganiello et al. describes the metabolomic analysis of axenic cultures of an endophytic Fusarium isolate (A1021B), formerly ascribed to the FIESC Fusarium species complex. Metabolomic data and molecular analysis of barcoding regions indicate that A1021B belongs to F. babinda, rather than the FIESC complex.

Minor Spelling/ Phrase:

Line 59: Please check phrase. E.g. Growing the fungus in ....(CDB) liquid culture fusarinol, .......

Line 79: CDB: please give supplier

Line 91: remove hyphen between 10 and g

Line 101: check phrase.

Line 159/160: check phrase: ....PDB was less .....

Same in 198 replace resulted by: were

L 206: ...at both time points (delete: the)

L 207/208...CBD was the least inductive medium...

L278: please specify : non-conforming strain: What do you mean by this?

Figure 1: increase font size in A. Avoid overlapping of writing and dots.  A and B are missing in figure. In A a legend for what is red/green/blue would be nice.

Figure 2: Increase size of numbers in circles. A and B are missing.

Major:

Line 150 ff: F. concolor not a good outgroup since it is relatively closely related to A1021B. The accession numbers MK984207, ..06, and ..08  are not publicly available in gene bank. They need to be uploaded for the further review and publication process.

Line 227: Under which axenic culture conditions are trichothecenes in other Fusarium strains produced?

Please clarify why you show two phylogenetic trees and not just one for the concatamer of all 4 marker genes. Also please include in the new phylogenetic analysis a higher number of Fusarium species. The ITS seq Blast e.g. shows high identity to F. acaciae-mearnsii and F. tricinctum.

Please provide a phylogenetic analysis tree, which shows the length of the branches.

Line 310: The data support that A1021B is more closely related to F. babinda than to F. concolor  and FIESC. A tree with branches showing substitutions and a broader species range will be more informative.

Author Response

Thank you for your valuable comments. We revised our manuscript according to the below notes.

Line 59: Please check phrase. E.g. Growing the fungus in ....(CDB) liquid culture fusarinol, .......

The sentence was changed to: “Fusarinol, a derivative of fusaric acid (FA), was the main extrolite purified from cultures grown in Czapek-Dox broth (CDB) [25]”.

Line 79: CDB: please give supplier

done

Line 91: remove hyphen between 10 and g

done

Line 101: check phrase.

The sentence was changed to: “Mobile phases consisted in (A) water (Cromasolv® Plus, LC-MS-Sigma) and (B) acetonitrile (Cromasolv® Plus, LC-MS-Sigma) both acidified with 0.1% LC-MS grade formic acid”.

Line 159/160: check phrase: ....PDB was less .....

done

Same in 198 replace resulted by: were

done

L 206: ...at both time points (delete: the)

done

L 207/208...CBD was the least inductive medium...

done

L278: please specify : non-conforming strain: What do you mean by this?

We refer to single isolates found in the course of investigations carried out in natural contexts which cannot be ascribed to any known species.

Figure 1: increase font size in A. Avoid overlapping of writing and dots.  A and B are missing in figure. In A a legend for what is red/green/blue would be nice.

done

Figure 2: Increase size of numbers in circles. A and B are missing.

done

Line 150 ff: F. concolor not a good outgroup since it is relatively closely related to A1021B. The accession numbers MK984207, ..06, and ..08  are not publicly available in gene bank. They need to be uploaded for the further review and publication process.
F. concolor was used as outgroup based on the work by O'Donnell et al. 2009. We moved ref. [41] to line 221 to make this clearer. Accessions in GenBank are publicly available after a few weeks. Sequence MK968883 is already available, while the rest will be available in the next days. For any necessity concerning the reviewing process, we attach the other sequences.

>Seq2 [organism=Fusarium babinda] [strain=A1021B] translation elongation factor (TEF1) gene AAGACTCACCTTAACGTCGTCGTCATCGGCCACGTCGACTCTGGCAAGTCGACCACTGTGAGTACTACCCTCGACGATCTGCTTCTTTGCACTCGTCAATCTCGCCTTAGATATGGCGGGGTATGCCTCAAAACGCAACATGCTGACATCCTTTAACAGACCGGTCACTTGATCTACCAGTGCGGTGGTATCGACAAGCGAACCATCGAGAAGTTCGAGAAGGTTAGTCACTTGCCCTTCGATCGCGCGCCCTTTTGCCCGTCGAGTTCCCTTTCGAATCACTCCCATACGACTCGATCAGCGCCGGATACCCCGCTTGAGTCCAAAAATTTTGCGGTGCGACCGTTAATTTTTTTGGTGGGGTATCTACCCCGCCACTCGAGTGACGGGCGCGCTTGCCCTGTTCCCACAAAATCATCATAATGGGCGCGCATCATCACGTGTCAATCAGTCACTAACCATTTGATAATAGGAAGCCGCTGAGCTCGGTAAGGGTTCCTTCAAGTACGCTTGGGTTCTTGACAAGCTCAAAGCCGAGCGTGAGCGTGGTATCACCATCGATATTGCTCTCTGGAAGTTCGAGACTCCTCGCTACTATGTCACCGTCATTGGTATGTTGCCACTGTTACTGTCACCTTAGTCTTGATCTCATGCTAACATCTCATTCAGACGCTCCCGGTCATCGTGACTTCATCAAGAACATGA

>Seq3 [organism=Fusarium babinda] [strain=A1021B] calmodulin (CAL1) gene

CTCCGTCTTTTGTTGGCTTGGCCTTGCTTGCAGTTGTCGCTAACCTGTTTGTGTAGGACAAGGATGGTGATGGTGAGTGATACTCCCCTCGCGATGTTTCTTTGCTAGCCCAGCACGAAACCCAAATCGACCGCAACAAAGCATCGAATAACTACAAATCTTTGCATCCACCTCCCTTCGATATTGATTAATCGGAAACATGAGCTAAACGCTTCACTATAGGCCAGATTACCACCAAGGAGCTCGGTACTGTCATGCGCTCTCTTGGCCAGAACCCCTCCGAGTCCGAACTTCAGGACATGATCAACGAGGTTGACGCTGACAACAACGGCACCATCGACTTCCCTGGTGCGTAACTTCTCAAGACGGTTGAAGGACGCCGTGCTAACAATTGAGCAAAGAGTTCCTTACCATGATGGCGCGCAAGATGAAGGATACCGACTCTGAGGAGGAGATCCGCGAGGCTTTCAAGGTGTTCGACCGTGACAACAACGGCTTCATTTCTGCTGCCGAGCTTCGACACGTCATGACCTCTATCGGCGAGAAGCTCACCGACGATGAGGTTGATGAGATGATCCGAGAGGCTGACCAGGACGGTGATGGCCGAATCGACTGTGAGTGACTTGAGATTGGGTAATACAACACCAACCATGCACCTTTACTAACAGAATTTATAGACAACGAGTTCGTCCAACTCTGGGTG

>Seq4 [organism=Fusarium babinda] [strain=A1021B] ß-tubulin (TUB2) gene

CTTGCACGGCCTCGACAGCAATGGTGTTTACAACGGTACCTCCGAGCTCCAGCTCGAGCGTATGAGTGTCTACTTCAACGAAGTATGTTTCACCAGCCCATTTCTATGAATCCCATGCTCACACACTTAGGCCTCTGGTAACAAGTATGTTCCCCGTGCCGTCCTCGTCGATCTCGAGCCTGGTACCATGGACGCCGTCCGTGCCGGTCCTTTCGGTCAGCTCTTCCGCCCCGACAACTTTGTTTTCGGTCAGTCCGGTGCCGGAAACAACTGGGCCAAAGGGTCACTCACC

Line 227: Under which axenic culture conditions are trichothecenes in other Fusarium strains produced?
Most Fusarium strains readily produce trichothecenes on maize kernels, as implied by statement at line 158.

Please clarify why you show two phylogenetic trees and not just one for the concatamer of all 4 marker genes. Also please include in the new phylogenetic analysis a higher number of Fusarium species. The ITS seq Blast e.g. shows high identity to F. acaciae-mearnsii and F. tricinctum.
The two trees are shown for documenting how the phylogenetic position of A1021B changed after the deposit in GenBank of sequences concerning F. babinda. We did not consider to add more species since our purpose was just to show relatedness of A1021B with FIESC, F. concolor and F. babinda; moreover ITS sequences proved to be unreliable in the identification of this strain (reff. 24-25).

Please provide a phylogenetic analysis tree, which shows the length of the branches.

We chose the circled representation for graphic reasons. In this form Mega7 does not allow to report the length of the branches. We would prefer to maintain this graphic form, which probably is more eye-catching. However, if strictly necessary, we are available to introduce the requested modification.
Line 310: The data support that A1021B is more closely related to F. babinda than to F. concolor  and FIESC. A tree with branches showing substitutions and a broader species range will be more informative.

Again, our purpose was just to show relatedness of A1021B with FIESC, F. concolor and F. babinda.  

Reviewer 2 Report

This article may partially support your controversy on bioactivity SMs and taxonomy:

http://dx.doi.org/10.1016/j.funbio.2013.05.007

line 144: Which model of molecular evolution did you use?

line 233 - Did you BLAST search CAL1 seqs? I would cross examine phylogenetic relations using Bayesian model with same sequences used here.

Line 242 – consider delete this line

Line 245 – it would have been of great help if you would have kept all previous seqs and added for the new inference the F. babinda ones…Otherwise you may keep the initial phylogenetic tree as supplementary material and/or only to mention the most important results (i.e. F. concolor and your strain). Should we understand that first phylo analysis was done with combined TEF and CAL and the second one only with TEF?

Line 249 – add in legend that you used only TEF1 seqs (which differ from the first analysis). More, as you should have understood from the first analysis, F. concolor cannot be used as outgroup, therefore polarization has occurred. It would be definitely be convenient and correct to include one or two outgroups (besides F. concolor, which is clearly not a suitable one) – a gradient phylo relation between them...

Line 310 – instead of “identified” -> lead/conduce, etc to identify…as…

Congratulations!

Author Response

Thank you for your valuable comments. We revised our manuscript according to the below notes.

This article may partially support your controversy on bioactivity SMs and taxonomy:

http://dx.doi.org/10.1016/j.funbio.2013.05.007
We do agree with the point of view proposed in this article. However, our aim is not to enter into the merits of a controversy on the sense of Linnean nomenclature as applied to fungi.

line 144: Which model of molecular evolution did you use?
The trees are based on the Tamura-Nei model.

line 233 - Did you BLAST search CAL1 seqs? I would cross examine phylogenetic relations using Bayesian model with same sequences used here.

Yes, but CAL1 sequence by itself was not informative for our purposes. 
Line 242 – consider delete this line
We think this line is relevant because it describes what seemed to result after a phylogenetic analysis carried out without F. babinda sequences, which were not available up to October 2018.

Line 245 – it would have been of great help if you would have kept all previous seqs and added for the new inference the F. babinda ones…Otherwise you may keep the initial phylogenetic tree as supplementary material and/or only to mention the most important results (i.e. F. concolor and your strain). Should we understand that first phylo analysis was done with combined TEF and CAL and the second one only with TEF?
As better pointed out in captions of figures 3 and 4, the first analysis was carried out on concatamers and the second only on TEF1 sequences. This is because, unlike TEF1 sequences, just a single sequence of CAL from F. babinda was available from Genbank, and TEF1 by itself can be informative enough (as an example, see ref. 23).

Line 249 – add in legend that you used only TEF1 seqs (which differ from the first analysis). More, as you should have understood from the first analysis, F. concolor cannot be used as outgroup, therefore polarization has occurred. It would be definitely be convenient and correct to include one or two outgroups (besides F. concolor, which is clearly not a suitable one) – a gradient phylo relation between them...
We specified that we used TEF1 sequences in caption of figure 4. In this figure F. concolor is not considered as outgroup  but as one of the species to be separated from F. babinda on account of the previous analysis.

Line 310 – instead of “identified” -> lead/conduce, etc to identify…as…

'identified' was changed to 'lead us to identify'.
Congratulations!
Thank you for Congratulations!